# Effect of COVID-19 Lockdowns on Eye Emergency Department, Increasing Prevalence of Uveitis and Optic Neuritis in the COVID-19 Era

**DOI:** 10.3390/healthcare10081422

**Published:** 2022-07-29

**Authors:** Joanna Przybek-Skrzypecka, Alina Szewczuk, Anna Kamińska, Janusz Skrzypecki, Aleksandra Pyziak-Skupień, Jacek Paweł Szaflik

**Affiliations:** 1Department of Ophthalmology, The Medical University of Warsaw, 61 Żwirki i Wigury Street, 02-091 Warsaw, Poland; anna.kaminska1@wum.edu.pl (A.K.); jszaflik@wum.edu.pl (J.P.S.); 2SPKSO Ophthalmic University Hospital, 24/26 Marszałkowska Street, 00-576 Warsaw, Poland; alina.yarashevich@gmail.com (A.S.); jskrzypecki@wum.edu.pl (J.S.); 3Department of Experimental Physiology and Pathophysiology, The Medical University of Warsaw, 3C Pawińskiego Street, 02-106 Warsaw, Poland; 4Department of Children’s Diabetology, Faculty of Medical Sciences in Katowice, Medical University of Silesia, 16 Medyków Street, 40-752 Katowice, Poland; aleksandra.ewa.pyziak@gmail.com

**Keywords:** COVID-19, ocular emergency, incidence, epidemiology, health services, conjunctivitis, optic neuritis

## Abstract

Background: The COVID-19 pandemic led to the reorganization of the health care system. A decline in health- and life-saving procedures has been reported in various medical specialties. However, data on ophthalmic emergencies during lockdowns is limited. Methods: We conducted a retrospective, observational, case-control study of 2351 patients registered at the ophthalmic emergency department of a tertiary hospital in Poland during three national COVID-19 lockdowns (March/April 2020, November 2020, and March/April 2021) and corresponding months in 2019. Results: The total number of visits declined from a mean of 720/month in the non-COVID era to 304/month during COVID-19 lockdowns (*p* < 0.001). Ocular trauma incidence dropped significantly from 2019 (non-COVID months) to 2020/2021 (COVID group mean 201 vs. 97 patients monthly, respectively, *p* = 0.03). Of note, the percentage of foreign bodies removal was significantly higher during lockdowns than corresponding time in the non-COVID era. A downward trend for vitreous detachment and macular disorders cases was observed between COVID and non-COVID time. Uveitis and optic neuritis patients were seen more often during lockdowns (*p* < 0.001 and *p* = 0.0013, respectively). In contrast, the frequency of conjunctivitis and keratitis, potentially COVID-related problems, decreased significantly in COVID-19 time (mean 138 vs. 23 per month in non-COVID vs. COVID lockdowns, respectively, *p* < 0.001). Conclusions: The overall number of eye emergency visits declined during COVID-19 lockdowns. Conjunctivitis and keratitis rates dropped during the lockdowns. Interestingly, the frequency of immune-mediated ocular conditions (uveitis, optic neuritis) increased significantly which might be triggered by SARS-CoV2 infection.

## 1. Introduction

Severe acute respiratory syndrome coronavirus 2 (SARS-CoV2) infection, triggering COronaVIrus Disease 2019 (COVID-19), emerged in Wuhan, China, in December 2019 [1]. Since then, over 570 million people worldwide have been infected and over 6.4 million died due to COVID-19 [2]. However, COVID-19 and its mortality rate are not the only problems that health care providers face during the pandemic. On the one hand, acute SARS-CoV-2 infection requires additional health care resources due to an increased number of emergencies (systemic viral illness), as well as chronic post-COVID syndromes (thrombosis- and chronic inflammation) [3]. On the other hand, a shortage of medical staff and equipment impaired effective care of well-known diseases, e.g., myocardial infarction or stroke [4,5]. Social distancing policies, as well as overcrowded emergency departments, might be responsible for the decreased number of patients seeking medical consultations and life-saving interventions. *Diegoli* et al., report that stroke- admissions declined by 36.4%, while life-saving interventions by about 40% [6,7,8].

Although various medical specialties have reported a decline in emergency procedures, ophthalmology did not follow, and key opinion leaders focused mainly on a significant decrease in elective procedures [9]. Indeed, elective procedures are the mainstay of ophthalmology, nevertheless delayed treatment of trauma, infection, and retinal or neuro-ophthalmic disorders is responsible for a significant percentage of vision-related disability [10]. Some of them might be successfully diagnosed and safely monitored via telemedicine methods as a substitute to direct ophthalmic care [11,12]. Prior to COVID-19 pandemic retinopathy of prematurity, diabetic retinopathy, and age-related macular degeneration were the mainstay of remotely controlled disorders in ophthalmology [13,14,15]. However, both the range of telemedicine tools (programs, devices) and the number of centers and doctors engaged in tele-visits, contributed to its development and providing care to patients with a broader spectrum of ocular disorders [16,17,18]. Nevertheless, some of the ocular disorders might not be properly diagnosed virtually, retina and choroid inflammations being the typical examples [12,17,19]. Additionally, considering an increasing number of reports on the autoimmune background of various post-Covid syndromes, it might be of special interest to investigate patterns of emergency eye consultations related to optic neuritis and uveitis [20,21].

Here, we wanted to study whether COVID-19 changed patterns observed in the emergency eye department. Our report is primarily focused on benign ocular diseases potentially linked to COVID-19 as well as eye-threatening disorders (including trauma). Finally, we looked at immune-mediated conditions which might be triggered by SARS-CoV-2 infection.

## 2. Materials and Methods

### 2.1. Methods

This study was approved by the Institutional Review Board of the Medical University of Warsaw, Poland. This study adhered to the tenets of the Declaration of Helsinki and was conducted as per the Standards for Reporting of Diagnostic Accuracy studies.

This is a retrospective, observational clinical study that included all patients seeking treatment at the Emergency Department at the Department of Ophthalmology, the Medical University of Warsaw, Poland, in three COVID-19 national lockdowns: (1) 16th of March to 15th of April 2020, (2) 1st to 30th November 2020, (3) 16th of March to 15th of April 2021. Compared to two control groups matched to the season’s outbreak: (1) 16th of March to 15th of April 2019 and (2) 1st to 30th November 2019. The dates reflected national lockdowns announced by the Polish Government. All lockdowns were further called “COVID-19” or “COVID” group and reference group— “non-COVID” group. The registry of patients was started in March 2020 by two ophthalmologists (J.P-S., A.S.) to monitor the epidemiology of ocular diseases at the largest Department of Ophthalmology in Poland, the Medical University of Warsaw, Ophthalmic Teaching Public Hospital in Warsaw (SPKSO) during SARS-CoV-2 pandemic.

We collected socio-demographic data: age, sex as well as ocular and systemic medical history. Additionally, we analyzed clinical diagnosis, time spent in the emergency department, number of “day” vs. “night” consultations (day defined as 6.01 am till 9.59 pm; night defined as 10 pm till 6.00 am) as well as number of hospital admissions. It is of note that we did not include emergency eye department diagnoses with a prevalence of less than 1% in either group.

Furthermore, we compared the prevalence of immune-mediated diseases, potentially COVID-19-mediated—(uveitis, optic neuritis) between the lockdowns and the corresponding time in 2019. We also checked vaccination status in patients suffering from immune-mediated diseases in the third lockdown group (the only studied group after the release of vaccination in Poland in January 2021).

### 2.2. Statistical Methods

Statistical analysis was performed with Statistica 13.1 (Tulsa, OK, USA). Data included in the study were analyzed with the T-student test (means for continuous variables when data was normally distributed, checked with the Shapiro-Wilk test) or the Mann-Whitney U test (for the incontinuous parameters). Nominal variables were presented as *n* (% of the group). Proportions for categorical variables were compared using the χ^2^ and Fisher exact test as appropriate. *P*-values less than 0.05 were considered statistically significant.

## 3. Results

### 3.1. Number of Patients

The study included 2351 patients consulted at the Emergency Eye Department at the Department of Ophthalmology, the Medical University of Warsaw during three COVID-19 lockdowns: (1) 16th of March to 15th of April 2020, (2) 16th of March to 15th of April 2021, (3) November 2020 and corresponding non-COVID days of the previous year: (4) 16th of March to 15th of April 2019, (5) November 2019. The total number of patients seen during COVID was 911 (3 lockdowns (90 days)) and 1440 in the non-COVID timeframe: 1440 (2 months, 60 days), with a mean of 10 and 24 patients per 24 h on-call, respectively (*p* < 0.001).

The number of emergency department admissions declined by 69% in the first lockdown in 2020 compared to the respective time in 2019. In particular, we noted an upward trend in the number of emergency visits between lockdowns. A reduction of 54% was observed for the second COVID-19 lockdown in November 2020 compared to November 2019 (Table 1). Socio-demographic data as well as ocular and systemic history, are presented in Table 1.

### 3.2. Time to Discharge

Next, we analyzed the technical aspects of visits to the emergency department (Table 2). The most prominent difference between lockdowns and reference time was waiting time in the emergency department from registration time-point until discharge (median 144 min vs. 369 min, COVID vs. non-COVID group, respectively, *p* < 0.001). In all groups, patients chose to come more often during the daytime (*p* = 0.55). The admission rate did not differ significantly between the groups (2–4.7% of all cases).

### 3.3. Trauma

Traumatic history was ascertained in 31.9% of the COVID group vs. 27.9% of the non-COVID group (*p* = 0.03). Considering each lockdown, we noted: 99 injuries in the first (38.4% of patients), 70 in the second (24.9%) and 122 in the third (32.8%) (*p* < 0.001) while in non-COVID there were 248 cases (29.9%) in the first and 154 cases (25.3%) in second time studied (*p* = 0.037). The total reduction in traumatic cases exceeded 50% (from a mean of 201/per month in non-COVID times to 97 cases/per month in lockdowns, *p* < 0.001). Of note percentage of visits related to the foreign body increased during first lockdown (*n* = 67, 26%) compared to corresponding time in 2019 (*n* = 134, 16%, *p* < 0.001).

### 3.4. Anterior Segment Diseases

Further, we were interested to study the incidence and percentage of most common anterior segment ocular problems diagnosed in the emergency department during COVID and non-COVID time. Within ocular surface problems, conjunctivitis, keratitis and dry eye syndrome were noted less often during lockdowns. However, the percentage of patients presenting corneal ulcers increased in the COVID era, most prominently in the second lockdown (23 cases, 8.2%) versus its reference time in 2019 (29 cases, 4.8%). The incidence of conjunctivitis also dropped from 150 and 125 cases in non-COVID 1 and non-COVID-2, respectively, to 18 (7%), 22 (8%), 29 (8%) cases per month during 1st, 2nd and 3rd lockdown, respectively (*p* < 0.0001). A similar pattern was observed for hordeolum (8, 10 and 21 cases in 1st, 2nd and 3rd lockdown, respectively, vs. 42 and 30 cases in non-COVID 1 and non-COVID 2 groups) and dry eye syndrome (10, 6, 13 cases in consecutive lockdowns, 44 and 29 in non-COVID groups), (Table 3).

### 3.5. Posterior Segment and Immune-Mediated Diseases

Compared to the same part of the year in 2019, the prevalence of posterior segment diseases changed during lockdowns. The incidence of a retinal tear, retinal detachment, vitreous detachment, and macular disorders was reduced (Table 4). On the contrary, uveitis and optic neuritis were diagnosed more often in the COVID-19 era. We reported 38 cases of uveitis in COVID-19 (4.2%) and 17 in non-COVID era (1.2%). There were 2 cases of intermediate uveitis (5.3%, one in the 2nd and 1 in the 3rd lockdown) and 36 cases of anterior uveitis (95.3%) in the COVID group. Similarly, 2 patients suffered from intermediate uveal inflammation in the non-COVID group (one in the 2nd and 1 in the 3rd lockdown), and the remaining 15 patients were diagnosed with anterior uveitis (88%). Of note, the third lockdown brought 21 cases of severe uveitis which is almost 4 times the number in the corresponding month in 2019 (Table 5). Twenty patients suffered from anterior uveitis, one from intermediate uveitis. Less than one-third (six patients) were vaccinated prior to the ocular symptoms (three of them with two doses, another three with one dose of Comirnaty Pfizer vaccine, the only available those times). The rate of the inflammation of the optic nerve increased from eight cases in 2019 to 16 cases in 2020 (mean incidence 0.133/day in non-COVID to 0.233/day in lockdowns, *p* < 0.001). None of the patients with optic neuritis in the third lockdown got vaccinated. Figure 1 presents details on uveitis and optic neuritis in our cohort.

## 4. Discussion

To the best of our knowledge, this is the first study analyzing the impact of the COVID-19 pandemic, including all three lockdowns, on the care and prevalence of common conditions in the ophthalmic emergency department. We found that during lockdowns, the total number of ocular emergency consultations decreased in our department by a factor of 2.4. We also noted a rising tide of autoimmunological ocular issues during lockdowns.

Previously numerous specialties, including neurology and cardiology, reported a fall in their emergency consultations during the pandemic [6,8]. Ophthalmology followed the path. Consistently with previous studies, we observed a decrease in emergency eye consultations, but our results dominate. *Poyser* et al. showed a 53% reduction in the number of eye emergencies during the first COVID-19 lockdown in Great Britain while we proved a 69% reduction [22]. One of the probable reasons for that disparity lies in the seasonal difference in diseases. Our study consists of data collected in March/April and November while the British study included the first lockdown in spring only. Furthermore, care in ophthalmology departments in Poland varies and is less centralized compared to ophthalmic care in the UK. Thus, patients in Polish settings most likely chose non-tertiary eye emergency clinics, less crowded than ours. Another important observation was made about a significantly shorter time the patients spent in the emergency department in the COVID era. In our center, the number of patients seeking help in the emergency department decreased significantly in the COVID-19 era but the number of ophthalmologists working in the setting was the same in COVID-19 and non-COVID eras. We prioritized the emergency department even in the shortage of medical staff during the epicenter of the pandemic. Furthermore, all unnecessary conversations and interactions were limited during the lockdowns (with a limited number of accompanying people being of note). Additionally, we presume the staff was strictly devoted to the emergency department, not being simultaneously occupied by different elective procedures as it took place in the non-COVID era. In general, our results reflect global trends, but the mean time spent in our emergency department in the COVID era is longer than in English, Israeli and American ophthalmic studies [23,24]. However, the available literature refers mainly to the first lockdown, while our project comprises all three lockdowns from 2020 and 2021.

Although a decrease in the absolute number of consultations related to acute anterior segment disorders, i.e., conjunctivitis, keratitis or minor ocular trauma was reported, their prevalence did not change significantly [25]. Interestingly, although conjunctivitis is perceived as one of the typical COVID-19 symptoms (5–27% of all patients with COVID), we did not see an increase in consultations related to conjunctivitis in our cohort [26,27]. We hypothesize that the self-limiting nature of conjunctivitis might discourage patients from seeking medical help in overcrowded emergency departments. Part of them might also be hospitalized due to COVID-19 and seen by a specialist in in-patient service. Patients might also choose telemedical consultations or local ophthalmology offices instead of the emergency department [13]. Finally, home offices, as well as virtual classrooms, decreased the transmission rate of common viral infections associated with seasonal conjunctivitis.

Similarly, we observed a reduced number of ocular trauma. Literature provides limited data on that topic. *Pellegrini* et al. showed a 68.4% reduction in eye injuries during the first Italian lockdown (10th March to 10th April 2020) compared with 2019. *Samya-Ali* et al. reported comparable incidence but a higher percentage of trauma in emergency cases profile between COVID and non-COVID era (March–June 2020 vs. 2019) in the British cohort [28]. Presumably, limited outdoor activities influenced the rate of trauma. Within the whole group of emergencies, the prevalence of retinal detachment (RD) was studied most thoroughly. *Patel* et al. found both a lower incidence of RD and a higher rate of worse outcome features (macula-off RD, longer time from symptom to diagnosis, more PVR) [29]. *Franzolin* et al. confirmed fewer cases of RD but a higher rate of macula-off RD in the COVID era. A multicenter study from Northern England showed a similar observation [30]. Of note, a 14-day delay in RD surgery due to quarantine led to a worse outcome [31]. We observed an almost equal incidence of RD in COVID and non-COVID groups.

Interestingly, some ocular conditions were reported more often during the COVID-19 pandemic. *Shroff* et al. previously found an increase in severe endogenous endophthalmitis [32]. Here, we found an increased number of consultations related to optic neuritis and uveitis—200% and 350%, respectively. Single reports regarding inflammatory involvement of optic tract and myelitis in COVID-19 were published previously [33,34]. Additionally, *Benito-Pascal* provided evidence of panuveitis with optic neuritis as an initial manifestation of the disease [35]. Although it is attractive to speculate that immune-mediated diseases might occur more often in SARS-CoV-2-infected patients, big data to support this is unavailable. Liu et al. speculated that SARS-CoV-2 might activate host cells leading to autoagression ‘cross-reactivity with host cells’ [21]. *Dalakas* supports the idea by reviewing all Guillain-Barre cases associated with COVID-19 [36]. Other reports on the higher incidence of: Kawasaki-like disease, autoimmune necrotizing myositis, and encephalopathies: show potential links [36,37,38]. Moreover, single reports on retinal nerve fiber layer alterations in COVID-19 patients also suggest potential viral triggers for nerve inflammation [39]. Further studies are needed to clarify the potential link between SARS-CoV-2 infection and neuroinflammatory ocular diseases.

This study has certain limitations. Firstly, it is a single-center retrospective study. We lack data from the whole region as other ophthalmology units do not run their own database. Secondly, within an observational report, we are unable to define causative mechanisms of a diminished number of disorders and symptoms. Moreover, the data about current infection was lacking, as we did not test every patient at the emergency department. Finally, we miss data about the influence of SARS-CoV-2 vaccination on the profile of diagnoses during lockdowns as they were introduced in Poland at the end of our project (January 2021) and covered only the elderly population.

To date, our study remains the first investigation of the impact of all three COVID-19 lockdowns on the eye emergency department. Of high probability, the fourth Delta SARS-CoV-2 wave is to come and ophthalmology services should be prepared and alert to the key problems in covering eye emergencies during the pandemic. We believe COVID-19 patients should be informed of ocular red flags as prompt recognition and treatment of ocular disorders is essential for the preservation of vision. This study might pave the way for informational campaigns about symptoms that should be addressed to preserve vision. Additionally, potentially COVID-related ocular disorders need to be investigated and addressed.

## 5. Conclusions

The number of eye emergency visits during COVID-19 lockdowns decreased significantly. Despite the potential link between Sars-CoV-2 infection and anterior segment-related problems, particularly conjunctivitis, we observed a lower incidence of conjunctivitis, keratitis, dry eye syndrome, hordeolum, and subconjunctival hemorrhages. We presumed that social distance rules and fear of contracting COVID-19 discouraged people from seeking help at an emergency department. Additionally, we observed an increased prevalence of uveitis and optic neuritis during the lockdowns. However, further research is needed to clarify its potential link to SARS-CoV-2 infection, SARS-CoV-2 vaccinations or other origins.

## Figures and Tables

**Figure 1 healthcare-10-01422-f001:**
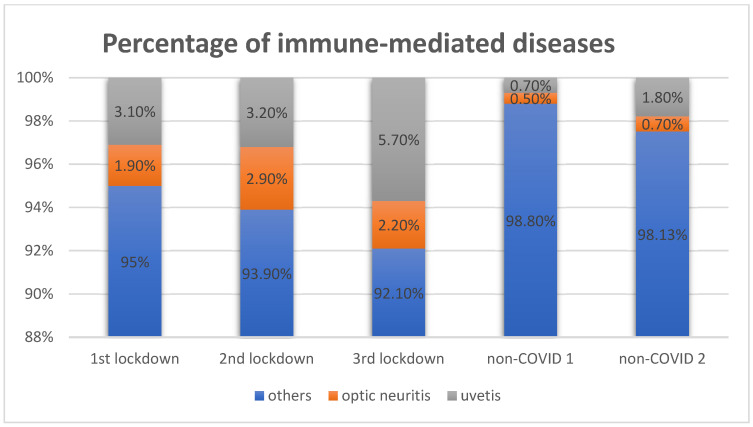
Proportion of immune-mediated diseases in the Emergency Eye Department, the Medical University of Warsaw in three lockdowns (1st lockdown = 16th of March to 15th of April 2020, 2nd lockdown = 1st to 30th November 2020, 3rd lockdown = 16th of March to 15th of April 2021 and corresponding time in 2019 non-COVID 1 = 16th of March to 15th of April 2019, non-COVID 2 = 1st to 30th November 2019).

**Table 1 healthcare-10-01422-t001:** Socio-demographic characteristics of patients admitted to the Emergency Department, Ophthalmology Unit, the Medical University of Warsaw, Poland, in three COVID-19 national lockdowns: 1st lockdown = 16th of March to 15th of April 2020, 2nd lockdown = 1st to 30th November 2020, 3rd lockdown = 16th of March to 15th of April 2021 and corresponding time in 2019 (non-COVID 1 = 16th of March to 15th of April 2019, non-COVID 2 = 1st to 30th November 2019).

	Whole Cohort	1st Lockdown	2nd Lockdown	3rd Lockdown	Non-COVID 1	Non-COVID 2	*p* Value
**Number of patients** **(% of the whole cohort)**	2351	258	281	372	830	610	*** *p* < 0.001**
	(100%)	(11%)	(12%)	(15.8%)	(35.3%)	(25.9%)	
**Sex:** **Male** ***n* (%)** **Female *n* (%)**	1146	92 (35.7%)	126 (44.8%)	175 (47%)	411 (49.5%)	342 (56.1%)	*** *p* < 0.001**
	1205	166 (64.3%)	155 (55.2%)	197 (53%)	419 (50.5%)	268 (43.9%)	**** *p* = 0.014**
**Age:** **Median (Q1:Q3)**	52	49	56	53	51	54	* *p* = 0.33
	35–68	36–66	39–70	36–69	34–67	35–68	** *p* = 0.17
**Chronic general disease** ** *n* ** **% of the group**	461	47	84	98	127	105	*** *p* < 0.0001**
	19.6%	18.2%	29.9%	26.3%	15.3%	17.2%	** *p* = 0.06
**Ocular chronic disease present** ** *n* ** **% of the group**	758	85	104	126	259	184	* *p* = 0.28
	32.2%	33%	37%	33.9%	31.2%	30.2%	** *p* = 0.57

* *p* whole cohort difference between 5 groups, chi-square test or ANOVA Kruskal-Wallis test. ** *p* difference between lockdowns, chi-square test.

**Table 2 healthcare-10-01422-t002:** Technical data on Emergency Department, Ophthalmology Unit, the Medical University of Warsaw in three lockdowns (1st lockdown = 16th of March to 15th of April 2020, 2nd lockdown = 1st to 30th November 2020, 3rd lockdown = 16th of March to 15th of April 2021and corresponding time in 2019 (non-COVID 1 = 16th of March to 15th of April 2019, non-COVID 2 = 1st to 30th November 2019).

	1st Lockdown	2nd Lockdown	3rd Lockdown	Non-COVID 1	Non-COVID 2	*p* Value
**Time spent in A&E until discharge, median** **(minutes)**	149	114	171	452	286	*** *p* < 0.001** **** *p* < 0.001**
**Part of the day** **Night *n* (% of the group)**	32 (12.4%)	23 (8.2%)	32 (8.6%)	104 (12.5%)	44 (7.2%)	*** *p* = 0.005** ** *p* = 0.18
**Admission to the hospital *n* (% of the group)**	12 (4.7%)	7 (2.5%)	11 (3%)	18 (2.2%)	12 (2%)	* *p* = 0.19 ** *p* = 0.33

* *p* difference between whole group, Kruskal-Wallis test or chi-square test. ** *p* difference between lockdowns, chi-square test. “Night” defined as 10 pm–6 am.

**Table 3 healthcare-10-01422-t003:** Prevalence of anterior segment and adnexae disorders in COVID lockdowns (16th of March–15th of April 2020, 2021, November 2020, 90 days) and non-COVID times (16th of March–15th of April, November 2019, 60 days), total prevalence and incidence per day and percentage of all diagnosis.

Clinical Diagnosis	Incidence per Month in COVID Lockdowns	Incidence per Month in Non-COVID Era	Part of the Whole Group COVID [%]	Part of the Whole Group Non-COVID [%]	*p* Value
**Conjunctivitis**	**23**	**138**	7.6%	19.1%	***p* = 0.0006**
**Keratitis**	**20**	**33**	6.6%	4.5%	***p* = 0.0079**
Dry eye	10	37	3.2%	5.1%	*p* = 0.728
Hordeolum	13	36	4.3%	5%	*p* = 0.5688
Subconjunctival hemorrhage	13	36	4.2%	4.9%	*p* = 0.5688

“*p*” assessed with chi-square test or Fisher test as appropriate.

**Table 4 healthcare-10-01422-t004:** Posterior segment diseases in COVID lockdowns (16th of March-15th of April 2020, 16th of March–15th of April 2021, November 2020, 90 days) and non-COVID times (16th of March–15th of April, November 2019, 60 days), total prevalence and percentage of all diagnoses.

	COVID	Non-COVID	COVID- Part of the Whole Group [%]	Non-COVID Part of the Whole Group [%]	*p* Value
Retinal detachment and tear	46	48	5.1%	3.3%	*p* = 0.254
Macular disorders	32	41	3.5%	2.9%	*p* = 0.991
Retinal vessel abnormalities	10	19	1.1%	1.3%	*p* = 0.296
Vitreous detachment	44	80	4.8%	5.6%	*p* = 0.276
Glaucoma	37	48	4.1%	3.3%	*p* = 0.959
**Optic neuritis**	**21**	**8**	**2.3%**	**0.6%**	***p* = 0.0013**

“*p*” COVID vs. non-COVID assessed with chi-square test or Fisher test.

**Table 5 healthcare-10-01422-t005:** Immune-mediated disorders with increasing prevalence in COVID-19 lockdowns (1st: 16th of March–15th of April 2019, 2nd: November 2020, 3rd: 16th of March–15th of April 2021) and controls: (non-COVID 1: 16th of March-15th of April 2019, non-COVID 2: November 2019).

Diagnosis	1st Lockdown *n* = 258	2nd Lockdown *n* = 281	3rd Lockdown *n* = 372	Non-COVID 1 *n* = 830	Non-COVID 2 *n* = 610	*p* Value
**Uvetis**	8	9	21	6	11	***p* < 0.0001**
**Optic** **neuritis**	5	8	8	4	4	***p* < 0.0001**

“*p*” assessed with chi-square test for the difference between the groups.

## Data Availability

All the data supporting the results are available upon request on Joanna Przybek-Skrzypecka’s e-mail address: joanna.przybek-skrzypecka@wum.edu.pl.

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
