# Peer review of "Effect of COVID-19 Lockdowns on Eye Emergency Department, Increasing Prevalence of Uveitis and Optic Neuritis in the COVID-19 Era"

_healthcare, 2022, doi:10.3390/healthcare10081422_

Round 1

Reviewer 1 Report

A study by Joanna Przybek-Skrzypecka et al.  entitled “Effect of COVID-19 lockdowns on eye emergency department. Are autoimmune eye disorders to come?” presents retrospective observational analysis of eye care at the emergency department (ED) in a single institution during the COVID-19 pandemic (three national lockdowns). The analysis is based on the impressive number of over 2300 patients who were seeking care due to eye emergency conditions. The authors focused on how pandemic affected the number of diagnoses and how they varied compared to non-pandemic time. Unsurprisingly, the main observation was that the number of visits at the ED declined during the pandemic period, as shown in the previous reports. Whereas the patients seeking due to conjunctivitis and keratitis decreased in number, the individuals with uveitis or optic nerve inflammations increased. The authors suggest this could be related to COVID-19 eye complications.

This study has few main points:

1.     The manuscript text is generally well-written; graphs and tables ale clear and well-presented, and support results consistently.  

2.     The three presented lockdown periods (each approximately one month in length) are well-compared to controlled periods without lockdown conditions and prior to pandemic, which represent appropriate study design.

3.     Even though this is a report from a single institution in one country, it provides data that could be useful for broader international community.

Weaknesses:

1.     Milder phrasing/language edits.

2.     Although this study has an impressive number of individuals included, this is the single centre report. Would be interesting to refer to the situation in other institutions worldwide.

3.     Some comments need to be addressed as mentioned below in Broad and Specific comments.

Broad comments:

1.     I would suggest changing the title of the manuscript (link the second statement more clearly/attractive to the topic).

2.     Introduction, 2nd paragraph: add information with relevant references about telemedicine and tele-visits as an important substitute to ophthalmic care during the pandemic.

3.     The readability and clarity of the manuscript would be improved by dividing the Results Section into subsections with subtitles summarizing cardinal findings in one sentence.

Specific comments:

1.     Introduction, row 5 (starting with “On the one hand,…” and “Both emergency..”): please edit these statements for better readability, merge two statements into one.

2.     Results, 1st paragraph: what was the number of patients included, over 2300 or 2500? There is some discrepancy in abstract and results section, table etc. Please correct.

3.     Results, 2nd paragraph: place “.” after (Table 1) and remove dot after 2019.

4.     Results, 2nd paragraph: change to “…and systemic history are presented in Table 1”.

5.     Table 1: how would you refer data on those patients who had chronic general or chronic ocular conditions to the observed increased numbers of uveitis or optic neuritis during the pandemic time? Have those patients with previously known chronic conditions contributed (significantly) to increase number of visits during the pandemic? Have those patients with inflammatory eye conditions been tested for COVID-19 or had evidence of past infection?

6.     Results, 3rd paragraph: how could you explain the length of ED visit during pandemic vs. non-pandemic condition? Please, write the reason in the manuscript text.

7.     Results, 5th paragraph: please improve this sentence “Table 3 and Table 4 show incidence and percentage of most common ocular problems diagnosed in emergency department in studied time divided into anterior and posterior segment diseases.” Start paragraph with description what was studied, e.g. “Further, we were interested to study….”. As in my comment #3 in Broad comments: add subtitle to this paragraph.

8.     Results, 5th paragraph, next sentence “Within ocular….”, replace colon with comma instead.

9.     Results, 5th paragraph, sentence starting with “Conjunctivitis’ incidence…” – please replace with “The incidence of conjunctivitis…”

10.  Discussion, 1st paragraph: change to “…on care and prevalence of common conditions in the ophthalmic emergency department”.

11.  Discussion, 2nd paragraph: change the phrase to “Consistently with the previous studies, we observed…”.

12.  Discussion, 2nd paragraph: improve this statement to e.g. “Care in ophthalmology departments in Poland varies  and is less centralised compared to ophthalmic care in the UK, thus patients in Polish settings most likely chose less crowded, non-tertiary eye emergency clinics than ours”.

13.  Discussion, 3rd paragraph: change “the ward” to “in-patients service”.

14.  Discussion, 5th paragraph: what type of uveitis was the most prevalent in your cohort? Grossly define: anterior, posterior, intermediate, panuveitis? Could you refer here more specifically questions from my Comment 5 (Specific comments)?

15.  So, the conclusion and the 2nd statement in the manuscript Title is a speculation. I suggest, the authors are careful with formulation conclusions and giving the initial title while these findings are not proved here.

16.  Conclusion, is “A&E department” an “emergency department”? Correct accordingly.

Author Response

Dear Reviewer,

we are very grateful for your detailed comments. We kindly appreciate your time and effort to evaluate and advise on how to improve our research paper. Please find the answers for your remarks below. They are divided accordingly to the groups of your revision: “broad comments”, “specific comments” and “weaknesses” of the study.

Sincerely,

Joanna Przybek-Skrzypecka, MD, PhD

Reviewer 2 Report

Dear Authors,

Congratulations for your work. This study is offering a very interesting analysis of the resegmentation of Ophthalmic Emergency Departments adressability influenced by the COVID times. For sure, the higher incidence of ocular immune-mediated diseases potentially triggered by SARS-CoV-2 infection is a fact which a lot of ophthalmologists all over the world faced off, but, even it's a sensitive item, We should know how many patients affected by these kind of diseases were vaccinated and to check some correlations with the number and type of the vaccine they had. I suggest you to use also these data in your paper in order to have a clearer idea about the etiology of these ocular inflammations.

Author Response

Dear Reviewer,

Thank you for your time and effort to review our manuscript “Effect of COVID-19 lockdowns on eye emergency department. Are autoimmune eye disorders to come?”. Please find our answers to your remarks in the attached file.

Sincerely,

Joanna Przybek-Skrzypecka, MD, PhD
